# Transposable Element Insertions into the *Escherichia coli* Polysialic Acid Gene Cluster Result in Resistance to the K1F Bacteriophage

Kathryn M. Styles,[a] Rebecca K. Locke,[b] Lauren A. Cowley,[b] Aidan T. Brown,[c] Antonia P. Sagona[a]

[a]School of Life Sciences, University of Warwick, Coventry, United Kingdom
[b]Milner Centre for Evolution, Department of Biology & Biochemistry, University of Bath, Bath, United Kingdom
[c]School of Physics and Astronomy, University of Edinburgh, Edinburgh, United Kingdom

**ABSTRACT** Reviewing the genetics underlying the arms race between bacteria and bacteriophages can offer an interesting insight into the development of bacterial resistance and phage co-evolution. This study shows how the natural development of resistances to the K1F bacteriophage, a phage which targets the K1 capsule of pathogenic *Escherichia coli*, can come about through insertion sequences (IS). Of the K1F resistant mutants isolated, two were of particular interest. The first of these showed full resistance to K1F and was found to have disruptions to *kpsE*, the product of which is involved in polysialic acid translocation. The second, after showing an initial susceptibility to K1F which then developed to full resistance, had disruptions to *neuC*, a gene involved in one of the early steps of polysialic acid biosynthesis. Both of these mutations came with a fitness cost and produced considerable phenotypic differences in the completeness and location of the K1 capsule when compared with the wild type. Sequential treatment of these two K1F resistant mutants with T7 resulted in the production of a variety of isolates, many of which showed a renewed susceptibility to K1F, indicating that these insertion sequence mutations are reversible, as well as one isolate that developed resistance to both phages.

**IMPORTANCE** Bacteriophages have many potential uses in industry and the clinical environment as an antibacterial control measure. One of their uses, phage therapy, is an appealing alternative to antibiotics due to their high specificity. However, as with the rise in antimicrobial resistance (AMR), it is critical to improve our understanding of how resistance develops against these viral agents. In the same way as bacteria will evolve and mutate antibiotic receptors so they can no longer be recognized, resistance to bacteriophages can come about via mutations to phage receptors, preventing phage binding and infection. We have shown that *Escherichia coli* will become resistant to the K1F bacteriophage via insertion element reshufflings causing null mutations to elements of the polysialic acid biosynthetic cluster. Exposure to the T7 bacteriophage then resulted in further changes in the position of these IS elements, further altering their resistance and sensitivity profiles.

**KEYWORDS** bacteriophage, resistance, transposable elements, insertion sequences, IS2, evolution, genomes, host resistance

Address correspondence to Antonia P. Sagona, A.Sagona@warwick.ac.uk.

The authors declare no conflict of interest.

Bacteriophages have many potential uses in industry and in the clinical environment as an antibacterial control measure (1). However, several hurdles need to be overcome for the use of bacteriophage-based disinfectants and phage therapy to become more widespread (2–4). In particular, as with the rise in antimicrobial resistance (AMR) (5), it will be critical to improve our understanding of how resistance develops against these viral agents to prolong the useful life span of any novel products or treatments.

 

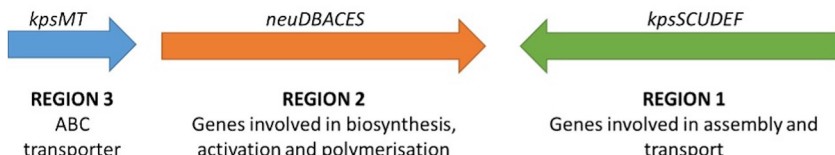

**FIG 1** Schematic of gene arrangement of the polysialic acid biosynthetic gene cluster. Arrows represent open reading frames and the direction of expression. Not to scale.

In the same way as bacteria will evolve and mutate antibiotic receptors, so they can no longer be recognized, resistance to bacteriophages can come about via the removal or alteration of phage receptors, preventing phage binding and infection.

EV36 is a non-pathogenic K-12-derived lab model strain containing the 14 genes of the polysialic acid biosynthetic cluster. The presence of the K1 capsule in *Escherichia coli* has long been known of as a virulence factor and is linked with strains that cause neonatal meningitis (6) and urinary tract infections (7). Not only does the K1 capsule provide a protective antiphagocytic barrier, but it is made from an $\alpha$-2,8-linked linear homopolymer of N-acetylneuraminic acid (sialic acid; NeuNAc), a polymer also found on the neuronal cell adhesion molecules (NCAM) of developing brains (8, 9). This structural similarity reduces the immunogenicity of K1-positive strains, allowing them to evade the immune system (10) and cross the blood brain barrier (9). The KIF bacteriophage uses the K1 antigen as its receptor, whereas T7 is normally blocked by the K1 capsule (10).

Transposable elements (TEs) or "jumping genes" will move and insert themselves throughout a genome via a "cut and paste" mechanism, thereby causing potential disruptions and null mutations to genes (11). There are several different types of TE. For example, transposons, (often associated with the transfer of AMR genes [12]) can be used to insert DNA or "genetic cargo" into a genome. Smaller insertion sequences (IS) contain only an open reading frame (ORF) for a transposase enzyme between two inverted terminal repeats, flanked by direct repeats, without any additional cargo. As the name suggests, the terminal repeats are the complements of each other and are recognized by the transposase and used to "locate" the transposable element to its new genomic location. The direct repeats are not part of the IS and are purely used to aid insertion, often being left as "footprints" in the genome (13, 14).

This study aims to offer a better understanding of the role of genetics in the competition and equilibrium between bacteria and bacteriophages, showing an insight into the development of bacterial resistances via the movement of IS through the bacterial genome. In this study, bacteriophage-resistant EV36 isolates were collected and characterized, through the investigation of relative growth and fitness, as well as a phenotypic analysis using confocal microscopy and the sequencing of entire genomes, to identify the causative genes in the development of phage resistance. We further investigated the K1F bacteriophage (15) and compared it with the T7 bacteriophage (10, 16) and how the *E. coli* strain EV36 (17) would respond to and develop resistance or sensitivities to these lytic phages. Specifically, we studied the natural development of resistance and then returned sensitivity, to the K1F bacteriophage, a phage which targets the K1 capsule of pathogenic *E. coli*. The genomic flexibility provided by IS movement appears to allow rapid adaptation to the presence versus absence of bacteriophages, removing potentially energy costly mutations once they are no longer needed.

## RESULTS

**Exposure to K1F bacteriophage produces K1F resistant isolates that are sensitive to T7.** We initially tested the ancestral EV36 wild type containing all 14 genes of the polysialic acid biosynthetic cluster intact (Fig. 1), which was previously unexposed to bacteriophages. We found that this wild-type strain was susceptible to K1F and showed a large area of clearance in a plaque assay when exposed to this phage, whereas it was largely resistant to T7 (Fig. 2A). Isolates of this strain were collected after

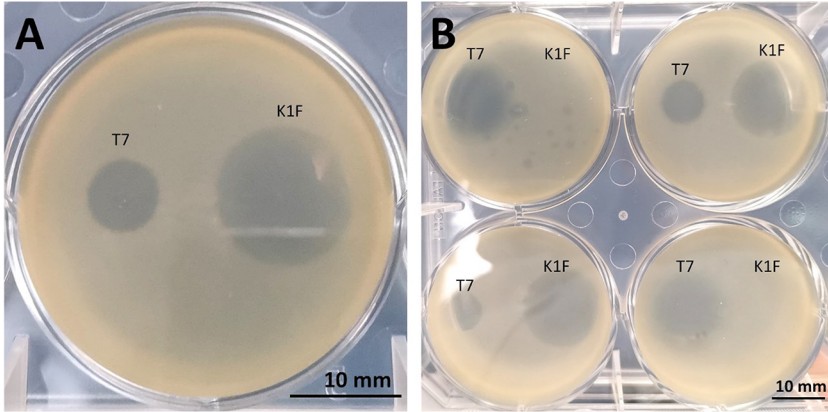

**FIG 2** Photographs of the phenotype of EV36 resistance and susceptibility to K1F and T7 bacteriophages. (A) Ancestral EV36 wild type, previously unexposed to K1F or T7. (B) Isolates of EV36 exposed to K1F for 20 h. Diameter of each well is 34.8 mm.

exposure to K1F (15) over 20 h. Some of these isolates showed plaques similar to those for the wild type, but some appeared to have acquired resistance to K1F, producing no plaques or only a very small plaque in its presence (Fig. 2B). The acquisition of resistance to K1F appeared to sensitize strains to T7 (8), as much larger plaques would form in the presence of T7.

**Resistance to bacteriophages comes with a fitness cost.** We selected two of the K1F-resistant isolates for further study: KMS2001 and KMS2005 (referred to as Mutant 1 and Mutant 5, Table 1). Mutant 1 showed no clearance when exposed to K1F in a plaque assay, whereas Mutant 5 showed a very small plaque ($\sim$1 to 2 mm in diameter). Mutants 1 and 5 were compared with the wild type using growth curves (Fig. 3) and the relative bacterial growth (RBG) rate (Table 2) was calculated for the mid-log phase (6 h). Although MG1655 (a K-12 strain that does not produce a PSA capsule) showed a RBG equivalent to that of EV36, both K1F-resistant mutants showed a significantly lower RBG compared to the wild type in the absence of phage ($P < 0.005$) (Table 2; Fig. 3A) indicating a fitness cost for resistance. However, both Mutant 1 and 5 had a significantly higher RBG in the presence of K1F when compared with the wild type in the presence of K1F ($P < 0.05$) (Fig. 3B).

Mutant 1 produced very similar growth curves, both in the presence and absence of K1F (Fig. 4A, light blue and yellow, respectively) and did not show a significantly different RBG when in the presence of K1F compared with its absence ($P > 0.05$, Table 1). Mutant 5 on the other hand appeared initially susceptible to K1F, but quickly developed resistance and showed a significantly lower RBG when exposed to K1F ($P < 0.05$).

**TABLE 1** List of *Escherichia coli* strains, bacteriophages, and plasmids used in this study

| Strain or plasmid | Alternative name | Extra information | Reference |
|---|---|---|---|
| EV36 | Wild type | *galP23 rpsL9* (*argA$^+$ rha$^+$ kps$^+$*), ancestral strain for the study, NCBI accession no.: CP079993 | 17 |
| KMS2001 | Mutant 1 | EV36 Δ*kpsE*, produced from ancestral EV36 exposed to K1F, NCBI accession no.: CP079992 | This work |
| KMS2005 | Mutant 5 | EV36 Δ*neuC*, produced from ancestral EV36 exposed to K1F, NCBI accession no.: CP079991 | This work |
| KMS2105 | | pSR647 in KMS2005 | This work |
| KMS2001a | Mutant 1a | Produced from KMS2001 exposed to T7 | This work |
| KMS2001b | Mutant 1b | Produced from KMS2001 exposed to T7 | This work |
| KMS2001a | Mutant 1c | Produced from KMS2001 exposed to T7 | This work |
| KMS2005a | Mutant 5a | Produced from KMS2005 exposed to T7 | This work |
| KMS2005b | Mutant 5b | Produced from KMS2005 exposed to T7 | This work |
| KMS2005c | Mutant 5c | Produced from KMS2005 exposed to T7 | This work |
| MG1655 | | F- lambda- *ilvG- rfb-50 rph-1*, K1 strain control (*kps*-) | 18 |
| Bacteriophage K1F | | | 8 |
| Bacteriophage T7 | | | 16 |
| pSR647 | | *neuC* in pCYB4 (an intein fusion plasmid from IMPACT-1 kit from New England Biolabs) | 24 |

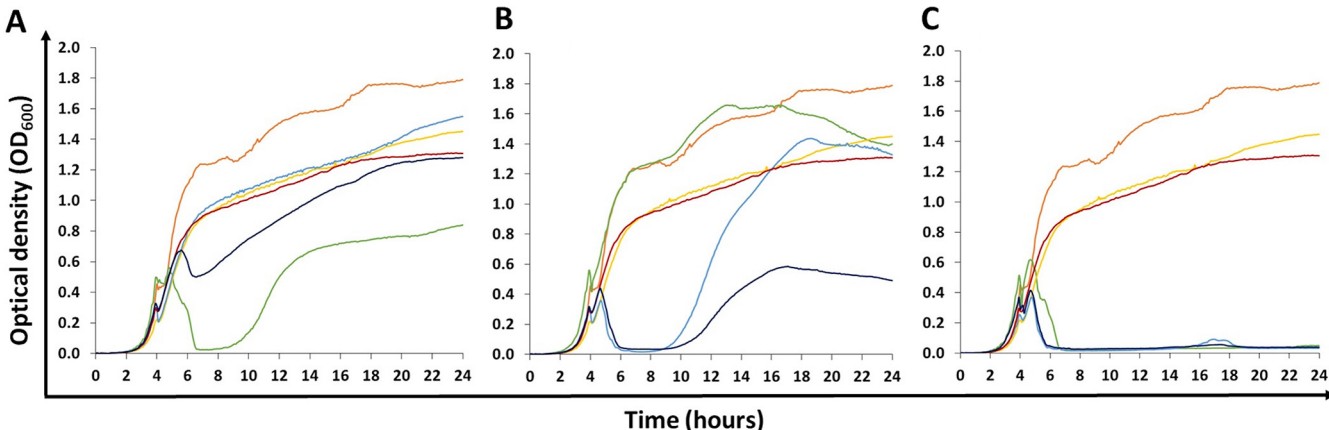

**FIG 3** Twenty-four-hour growth curves of EV36 strains and K1F resistant mutants both in the presence and absence of K1F and/or T7 bacteriophages. Orange, EV36 wild type; Green, EV36 + bacteriophage; Yellow, Mutant 1; Light blue, Mutant 1 + bacteriophage; Red, Mutant 5; Dark blue, Mutant 5 + bacteriophage. Bacteriophages used in each instance are denoted by different panels: (A) K1F, (B) T7, and (C) K1F and T7. Bacteriophages were added at the start of the log phase (4 h) to a final concentration of $1 \times 10^6$ PFU/mL ($5 \times 10^5$ PFU/mL of each phage for two-phage treatments).

Both mutants showed an initial susceptibility to T7, but Mutant 1 then appeared to develop full resistance, whereas Mutant 5 developed partial resistance (Table 1; Fig. 3B). The wild type, Mutant 1 and Mutant 5 were all susceptible to a combined treatment of K1F and T7 (Fig. 3C).

**Sequential phage exposure allows the production of mutants resistant to two phages.** We further exposed Mutants 1 and 5, which were previously isolated for their resistance to K1F, to the T7 bacteriophage for 20 h and six isolates resistant to this second bacteriophage were collected: KMS2001a-c and KMS2005a-c (referred to as Mutants 1a to c and 5a to c, Table 1). When grown in the absence of bacteriophages, growth curves produced with Mutant 5b had an RBG, which was equivalent to the wild type (97% ± 7%, $P > 0.05$) (Table 2) and the overall growth curve profiles and sensitivities seen for this mutant were also the most similar to the wild type (Fig. 4). All other mutants had a RBG below 80% of the wild type and showed different bacteriophage sensitivities compared to the ancestral stain. Upon gaining at least partial resistance to T7, Mutants 1a to c and 5a to c all showed a returned susceptibility to K1F, with one

**TABLE 2** Relative bacterial growth (RBG) and resistance profiles of isolates of EV36 mutants as revealed by 24-h growth curves[a]

| Strain | Relative bacterial growth (RBG) | Response to exposure to bacteriophages | | |
|---|---|---|---|---|
| | | K1F | T7 | Combination |
| EV36 | 1.00 ± 0.08 | S → PR | R → S | S |
| Mutant 1 | 0.69 ± 0.04 | R | S → R | S |
| Mutant 1a | 0.46 ± 0.30 | S | R | S |
| Mutant 1b | 0.52 ± 0.38 | S | R | S |
| Mutant 1c | 0.31 ± 0.25 | S | R | S |
| Mutant 5 | 0.73 ± 0.07 | PR → R | S → PR | S |
| Mutant 5a | 0.59 ± 0.10 | R | R | R |
| Mutant 5b | 0.97 ± 0.07 | S → PR | R → S | S |
| Mutant 5c | 0.79 ± 0.09 | S | PR | S |
| MG1655 | 0.94 ± 0.00 | R | S | S |

[a]RBG calculated during the mid-log phase (6 h) in the absence of bacteriophages. Classification of response to exposure to bacteriophages: sensitivity to bacteriophages is defined fully resistant (R; growth curve equivalent to the bacteria only control), partially resistant (PR; growth is above the baseline, but less than its control with no phage), or susceptible (S; defined as an $OD_{600}$ of, or close to, zero at some point in the study). Changes to sensitivities and the development of resistance or susceptibility are indicated using an arrow (→). The development of resistance is classified as developing full resistance (R; becomes equivalent to samples without phage) or partial resistance (PR; some resistance developed but does not return to levels of controls with no phage, e.g., as is seen for EV36 wild type with K1F). The development of susceptibility indicates a reduction in turbidity at later time points (usually 18 h onwards) when previously full resistance was seen. This was always only a small decrease.

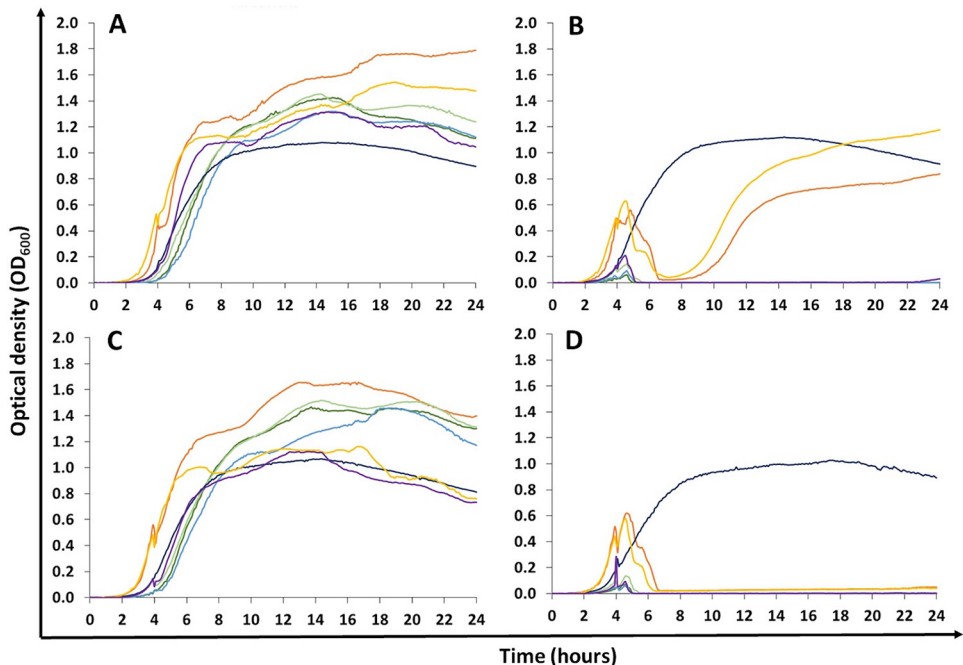

**FIG 4** Twenty-four-hour growth curves of EV36 strains and mutant strains produced by sequential exposure to K1F and then T7. Orange, EV36 wild type; Dark green, Mutant 1a; Light green, Mutant 1b; Light blue, Mutant 1c; Dark blue, Mutant 5a; Yellow, Mutant 5b; Purple, Mutant 5c. Bacteriophages used in each instance are denoted by different panels: (A) negative controls, no bacteriophages, (B) K1F, (C) T7, and (D) K1F and T7. Bacteriophages were added at the start of the log phase (4 h) to a final concentration of $1 \times 10^6$ PFU/mL ($5 \times 10^5$ PFU/mL of each for two-phage treatments).

exception: Mutant 5a, which appeared to be resistant to both K1F and T7 (Table 2; Fig. 4C and D), i.e., it appears to be a double resistance mutant. This did come with a fitness cost, however, with the RBG for Mutant 5a being less than 60% of the wild type.

**K1F resistant isolates show abnormal K1 capsule formation.** To understand in more detail the physiology of Mutants 1, 1a, 5, and 5a, which were isolated in the previous steps based on their resistance to K1F and their further exposure to T7, we stained these mutants for DNA and the polysialic acid capsule (PSA), and imaged them using confocal microscopy (Fig. 5). Mutants 1, 5, and 5a all gave phenotypes very distinct from the wild type, with < 1% of cells showing a wild-type phenotype (Table 3). Mutant 1 (K1F resistant and T7 susceptible) showed the presence of PSA, but this would accumulate at the poles of cells or was found spread through the cell but concentrated in isolated regions. Mutants 5 and 5a (at least partially resistant to K1F and T7) both showed no evidence of PSA production and a phenotype comparable to MG1655 (Fig. 5) (18). Mutant 1a (K1F susceptible and T7 resistant), on the other hand, showed a phenotype similar to the wildtype and appeared to have a complete PSA capsule. For the two K1-positive strains (EV36 and Mutant 1a) quantification revealed that in just over 96% of cells, there was evidence of a PSA capsule (Table 3).

**Long-read sequencing revealed disruptions to the PSA gene cluster.** To further validate these results and confirm whether the observed phenotype revealed by confocal microscopy corresponds to the genotype, the mutants were fully sequenced. Long-read sequencing of Mutants 1 and 5 (PacBio Platform, Novogene; NCBI accession numbers KMS2001: CP079992 and KMS2005: CP079991) revealed disruptions to the PSA gene cluster (Fig. 6). Mutant 1 was found to have a disruption in *kpsE* and Mutant 5 was found to have *neuC* disrupted. Long-read sequencing of Mutants 1a and 5a (PacBio Platform, Novogene, NCBI accession numbers: CP093368 and CP093369, respectively) revealed further changes to the PSA gene cluster (Fig. 6). The insertion into *kpsE* seen in Mutant 1 was found to be absent in the descendant, Mutant 1a. Mutant 5a, on the other

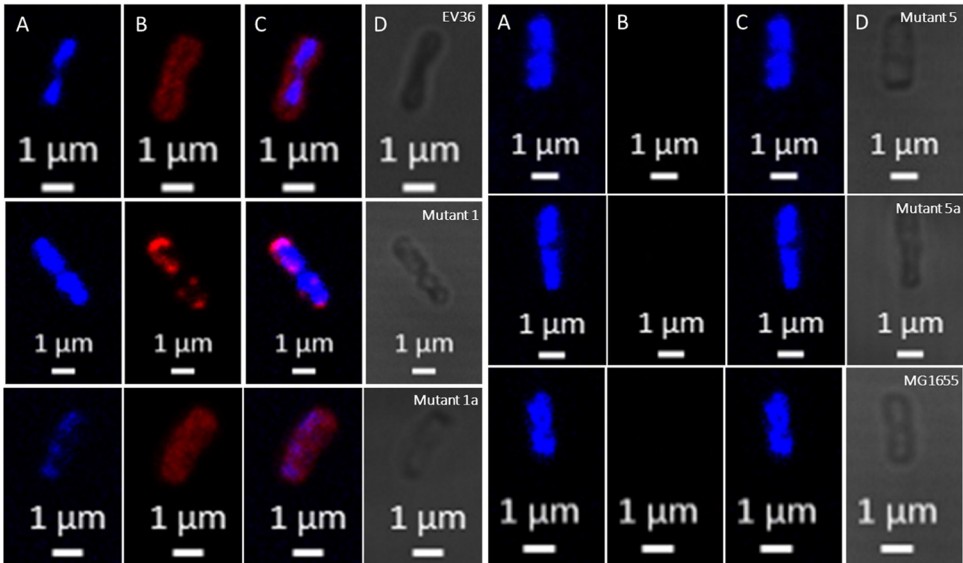

**FIG 5** Confocal microscopy images of the EV36 ancestral strain ($kps^+$ and $neuC^+$) compared with K1F and T7 resistant mutants and MG1655 ($kps$- and $neuC$-). For each strain there are four images of the same cell in different visualization modes: (A) DAPI DNA stain, blue, 725 gain; (B) polysialic acid capsule, red, 700 gain; (C) combined DNA and capsule image, red and blue; (D) bright field transmitted light only, gray.

hand, maintained the *neuC* disruption seen in Mutant 5. However, this mutant did have some additional genomic changes with a large section (~30kb) next to an IS5 element deleted (Fig. 6B). This deletion resulted in the removal of the gene for outer membrane porin PhoE.

These mutations appear to be a result of smaller IS (Fig. 7A), which contain only an ORF for a transposase enzyme between two inverted terminal repeats, flanked by direct repeats, without any additional cargo. The transposase binds to the end of an insertion sequence and catalyzes its cleavage and then movement to another part of the genome (Fig. 7B). Based on the sequencing results, the mutations in the PSA gene cluster were attributed to a transposase InsC for insertion element IS2D and an IS3-like element IS2 family transposase disrupting these genes. The large deletion that occurred in the double-phage-resistant Mutant 5a was attributed to an IS5 family transposase. No other single nucleotide polymorphisms (SNPs) or mutations were found in either mutant when compared with the wild type.

**(i) The addition of *neuC* returns PSA biosynthesis to mutants.** Finally, we performed a "rescue" experiment, in order to further confirm our sequencing data. In this "rescue" experiment, the addition of a plasmid containing *neuC* (pSR647), induced with IPTG, rescued the production of a complete PSA capsule in Mutant 5 (Δ*neuC*), as observed by confocal microscopy (Fig. 8). The capsule was visually comparable to the phenotype seen for the EV36 ancestral strain and Mutant 1a, confirming our previous results.

**TABLE 3** Quantification of cells observed via confocal microscopy that presented an abnormal versus normal polysialic acid capsule

| Sample type | Total cell count | Total abnormal cells | Relative values (%) | | T-test compared with EV36 ancestral strain |
|---|---|---|---|---|---|
| | | | Abnormal (without capsule) | Normal (with capsule) | |
| EV36 | 391 | 15 | 3.84% | 96.16% | 1.000 |
| Mutant 1 | 395 | 393 | 99.49% | 0.51% | <0.001 |
| Mutant 1a | 380 | 14 | 3.68% | 96.32% | 0.937 |
| Mutant 5 | 398 | 397 | 99.75% | 0.25% | <0.001 |
| Mutant 5a | 431 | 431 | 100.00% | 0.00% | <0.001 |

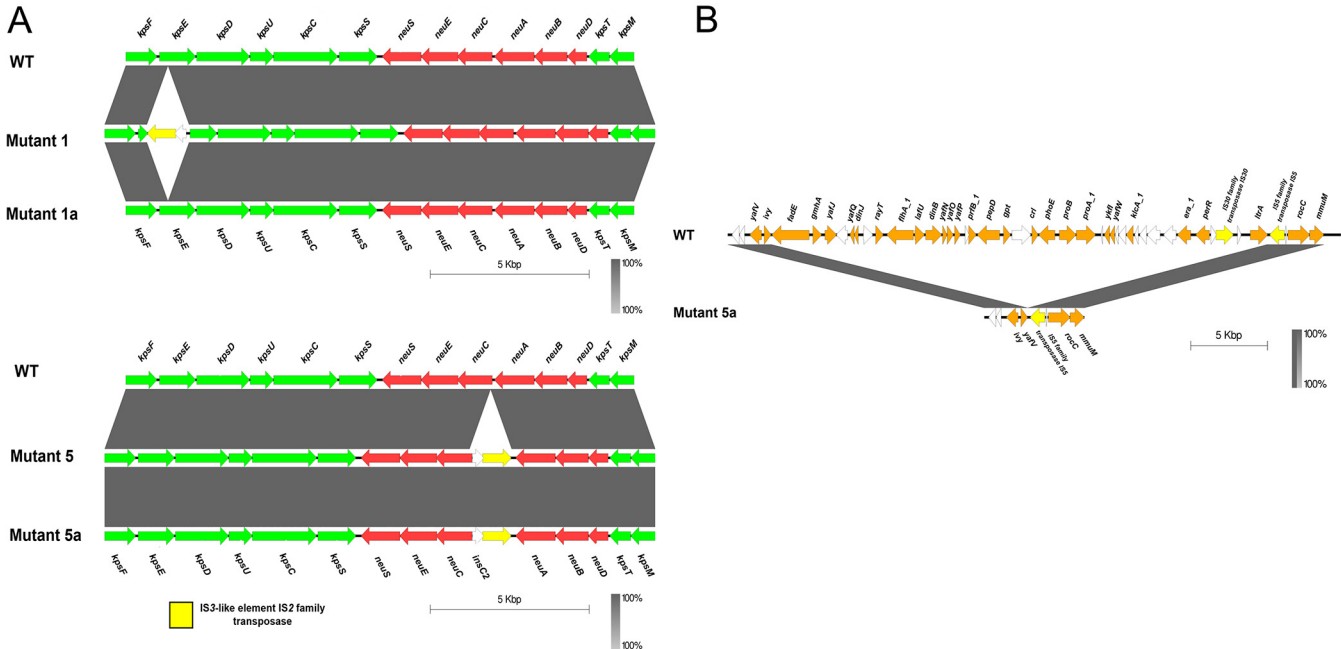

**FIG 6** BLAST comparisons of regions in mutants compared to wild-type *E. coli* EV36. Assemblies were generated with Pacific Biosciences (PacBio) sequencing technology. (A) Comparisons of the 17kbp polysialic gene cluster encoding the K1 capsid in mutants exposed to K1F bacteriophage compared with wild type. (Upper) Mutant 1 shows insertion sequence-mediated disruption in the *kpsE* gene whereas this is no longer present in Mutant 1a. (Lower) Mutant 5 shows disruption in part of the *neuC* gene, with this element still present in Mutant 5a. kps genes involved in polysialic acid transport to the cell surface are arranged in two regions: (i) *kpsMT* and (ii) *kpsSCUDEF* and are shown in green. *neuDBACES* genes responsible for K1 synthesis are shown in red. Transposase InsC for insertion element IS2D is shown in white and IS3-like element IS2 family transposase is shown in yellow. (B) BLAST comparison of a 16kbp region in wild-type to Mutant 5a, where a large ~30kbp segment adjacent to a IS5 element has been deleted, including outer membrane porin PhoE. Genes of unknown function are shown in white and insertion sequences in yellow. Arrows represent open reading frames and gray gradient shows BLASTn identity between regions. Figure produced with Easyfig v2.2.5 (42).

## DISCUSSION

The speed at which bacteria develop resistance to antibiotics has long been of concern. For example, the microbial evolution and growth arena (MEGA) plate study in 2016 showed that *E. coli* could colonize a plate containing a gradient which ended with 3,000 times their MIC of trimethroprim within just 12 days of inoculation (19).

In our study, all mutants collected were produced after only 20 h of exposure to K1F or T7 (no more than 60 replication cycles). It was found that resistant mutants were more readily created upon sequential exposure to K1F and then T7, rather than

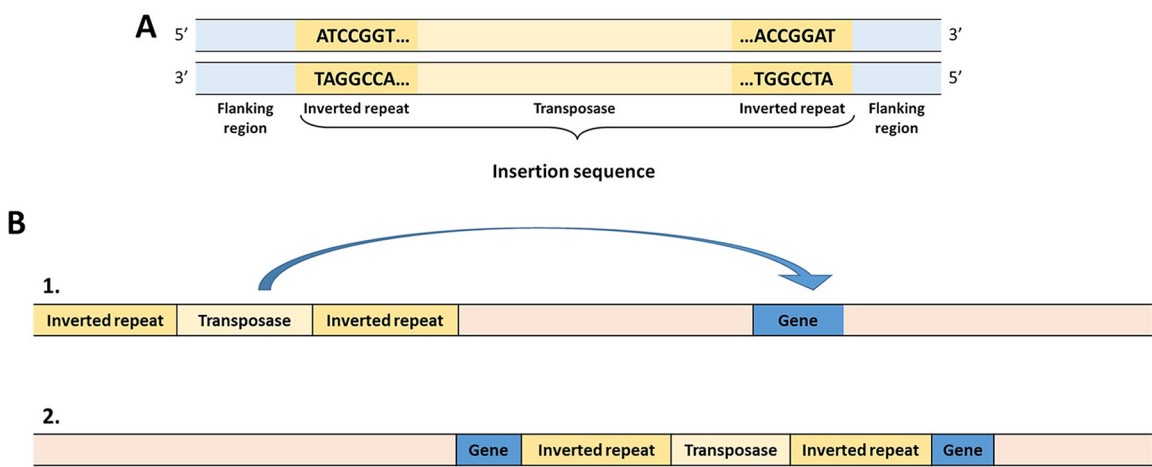

**FIG 7** (A) Schematic of an insertion sequence (IS). (B) Schematic of the mechanism of insertion sequence movement through a genome. Not to scale.

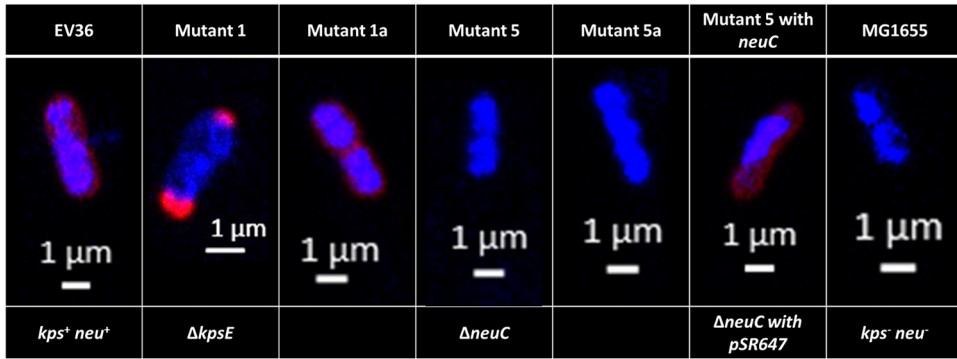

| EV36 | Mutant 1 | Mutant 1a | Mutant 5 | Mutant 5a | Mutant 5 with neuC | MG1655 |
|---|---|---|---|---|---|---|
| kps⁺ neu⁺ | ΔkpsE | | ΔneuC | | ΔneuC with pSR647 | kps⁻ neu⁻ |

**FIG 8** Confocal microscopy images of the EV36 ancestral strain (*kps*⁺ and *neuC*⁺) compared with K1F and T7 resistant mutants and "rescue" experiment Mutant 5 with pSR647 (containing *neuC*). All images are from a combined visualization of DNA (blue, DAPI, 725 gain) and the polysialic acid capsule (red, 700 gain) for a single cell.

simultaneous exposure, a phenomenon that has been reported in previous studies (20). Changes in susceptibilities to K1F or T7 were shown to usually come at a fitness cost to the host and were associated with a phenotypic change in the production and location of the K1 antigen, as seen via confocal microscopy. In particular, we investigated the mechanism through which resistance is developed, by undertaking long-read sequencing of two K1F resistant mutants and two of their T7 resistant progeny.

Long-read sequencing of Mutant 1 specifically revealed disruptions to the *kpsE* gene, the product of which is involved in polysialic acid translocation (21). Unsurprisingly, therefore, the phenotype observed for this mutant was one of PSA production, but in an unusual location, either accumulating at the poles of the cells or randomly concentrated throughout the cells. A similar phenotype was also observed by Vimr et al. in 1996, when *kpsE* knockout mutants were studied via electron microscopy (22, 23).

Mutant 5 did not show any K1 capsule when observed via confocal microscopy, and long-read sequencing showed that *neuC*, a gene involved in one of the early steps of polysialic acid biosynthesis (23), was disrupted in this mutant. Mutant 5 appeared to be slightly less resistant to K1F than Mutant 1 (particularly earlier on in growth), but it is unclear why disruptions to *neuC* provided less protection against K1F than disruptions to *kpsE*. Possibly alternative genes can produce homologues of NeuC and thereby can result in the production of a minimal level of polysialic acid, enough for some binding to K1F (although not enough to be detected microscopically). Earlier studies by Vann et al. showed that in *neuC* knockout mutants, PSA biosynthesis can be returned by the addition of exogenous sialic acid monomers or the synthetic return of *neuC* to a mutant using plasmid pSR647 (24), which we repeated in this study with the use of the same inducible plasmid.

Mutant 1a had a phenotype of PSA production, K1F susceptibility, and T7 resistance. This can be explained by the removal of the IS in the *kpsE* gene, thereby allowing the return of PSA biosynthesis. Mutant 5a showed a phenotype of no PSA production and resistance to both K1F and T7. Long-read sequencing of the genome revealed that this was due to the maintenance of the IS insertion in *neuC*. Mutant 5a had a deletion, which included outer membrane porin *phoE*, a known target of T7, which explains the phenotype change in T7 susceptibility (25).

Just under 4% of the two K1 positive strains (EV36 and Mutant 1a) were acapsular (lacked capsules, based on the polysialic acid capsule antibody used in this study) when observed with confocal microscopy. This is an interesting result and one to look at more thoroughly in future studies. This natural heterogeneity has been previously observed, and upregulation of the capsule has often been associated with challenges presented by the human cell environment (22, 26), where stochasticity is a fitness strategy to deal with a changing external environment (26). The selective pressure of the human cell environment was absent in this study, but the heterogeneity appears to

remain and the bacteria are still "ready for battle" in an environment that could change at any moment (27).

All but one of the mutants, which presented a pleiotropic effect of a beneficial mutation, had a mid-log RBG rate below 80% of the wild type, indicating that there is a fitness cost associated with the movement of transposable elements. Both Mutants 1 and 5 showed only single gene disruptions in the PSA cluster, with no other disruptions detected elsewhere in the genome and yet showed RBGs of 69% $\pm$ 4% and 73% $\pm$ 7%, respectively. MG1655 (kps-, neuC-) on the other hand, showed a RBG equivalent to EV36 indicating that production of the capsule is not significantly costly to the specific strain used in this study and that a lack of PSA is not detrimental to normal cell growth. It is possible therefore that we have selected for isolates with a greater rate of the movement of insertion sequences, which may have an associated metabolic cost. Alternatively, the mis-transport of PSA and the clearance of accumulated intermediates may impact on the fitness of the bacterium. Total elimination of the biosynthetic cluster (as in MG1655) may thus have a lower fitness cost than single gene mutations, as mutations within a non-essential gene cluster can lower fitness below that of carrying the entire functional cluster. Sequencing also only provided a snapshot of the location of ISs, but they have the potential to move again over the course of a growth curve, with potential effects on metabolism, growth and cell division, depending on which genes become disrupted. However, despite the fitness costs, both Mutant 1 and 5 had greater RBGs, compared with EV36, in the presence of K1F. In the selective environment of lytic bacteriophage exposure, the fitness costs associated with genetic plasticity and the movement of transposable elements appeared to be outweighed by the chance of survival if the target of the bacteriophage was altered so it could no longer be recognized (28, 29). When it comes to IS elements, EV36 could evolve to remove those, but the potential selection for phage resistance may have outweighed the costs. The IS elements potentially persist, not just because they increase the beneficial mutation rate, but also as selfish elements transmitted by conjugative plasmids, which, once acquired, are almost impossible to select out of the population.

Resistance was generated by transposable insertion sequences disrupting relevant genes for phage binding. In both first-generation K1F-resistant mutants that underwent long-read sequencing, there were two transposable elements (TEs) inserted into the PSA-biosynthetic cluster, leading to null mutations to the kpsE and neuC genes. These K1F resistant mutants were sequentially exposed to T7 and the resistant progeny were collected. Subsequent long-read sequencing of these second generation mutants revealed further movement of the TEs, with one mutant regaining a functional kpsE gene and with it, K1F susceptibility and the other maintaining the loss of the neuC gene, but losing the gene for outer membrane porin PhoE allowing this mutant to become resistant to both K1F and T7.

We identified a transposase InsC for insertion element IS2D and an IS3-like element IS2 family transposase as responsible for both observed disruptions to the PSA biosynthetic cluster in our resistant mutants, presenting a potential role for insertion sequences as a host defense mechanism in the evolutionary arms battle between bacteria and viruses (or other environmental challenges that require a rapid evolutionary response to survive) (30). So long as an insertion sequence does not interrupt an essential gene, the relocation of TEs can offer a distinct survival advantage (albeit with a potential fitness cost) to their host should the target of a bacteriophage (or even antibiotic) be altered so that it can no longer be recognized by the bacteriophage (or antibiotic). This type of change is also reversible, with the TEs able to move again, allowing a pathogen to return to "normal," with wild-type levels of fitness, after a threat has passed. Therefore, based on our results, resistance appeared to not only be fast, but also reversible. Although other mechanisms of developing resistance, e.g., horizontal gene transfer, may be much slower and require many more generation cycles, this ready propensity for bacteria to evade lysis by bacteriophages via evolution is concerning. One overarching benefit of phages, however, is their abundance and variety in nature

(31). Although resistance will develop, we have an almost infinite source of potential phage therapy options, which can be combined into multi-agent cocktails. As was shown in this study, and previously by Wright et al. in 2019 (20), the production of more successful resistant mutants are often created via sequential, rather than simultaneous exposure to more than one phage. The therapeutic use of simultaneously given phage cocktails therefore would be expected to largely prevent the resistances observed in this study. It has to be noted here that this study used EV36, rather than the wild-type *E. coli K1* strains. Therefore, due to the way EV36 was constructed, these mutations are not necessarily representative of potential mutations that could arise in wild-type *E. coli K1* strains. Previous studies with mutants of *E. coli K1* strains, with low level of the K1 capsule, have revealed loss of the cleaving activity of the endosialidase enzyme of the phage (32).

The IS elements that were inserted into the polysialic gene cluster in Mutants 1, 5, and 5a were found in seven additional locations in the EV36 genome. We therefore hypothesize that these IS elements have been duplicated, during cell replication, into a new location resulting in the observed phenotype change.

Several of the sequential mutants (not all of which were sequenced) showed a returned susceptibility to K1F upon gaining T7 resistance. Although the transposition rate of insertion sequences is greater than the excision rate (29, 33), it appeared that the insertions seen in Mutants 1 and 5 were excised and a genetical equivalent to the wild-type strain was restored. Indeed, Mutant 1a phenotypically showed the K1 capsule and sequencing revealed that the PSA cluster was again intact whereas its progenitor, Mutant 1, had disruptions to *kpsE*.

It should be noted that although this study does not represent the human cell environment, where there are temporal and spatial differences in phage-bacteria encounters, it does provide details on possible mechanisms by which bacteriophage resistance may develop, something which may be replicated in the *in vivo* context at a later date.

**Conclusions.** The presence of polysialic acid is critical to the pathogenicity of a range of *E. coli* strains. Here we have shown that insertion sequences causing disruptions to single genes in the PSA biosynthetic gene cluster produce mutants resistant to K1F, a phage known to target the K1 capsule. Mutations to the *neuC* and *kpsE* genes produced a phenotype showing no or abnormal K1-capsule production, respectively, and reduced relative bacteria growth compared with the EV36 wild type. We also showed that sequential exposure to K1F, and then T7, successfully resulted in the outgrowth of a mutant resistant to both phages (Mutant 5a), whereas simultaneous application of both phages resulted in very low growth rates due to the collapse of the bacterial population. It is advisable therefore to use cocktails of phages to target pathogens: bacteria have a range of mechanisms to evolve resistance to phages and we must therefore compete with this in the range of mechanisms we use to target them. Interestingly, sequential exposure to T7 and K1F phages also produced a mutant with an identical phenotype to the wild type (Mutant 5b). We suggest that the mechanism of mutation via insertion sequences is reversible and that this may constitute an evolutionary benefit of such mobile elements, namely, the ability to return to wild-type levels of fitness once an environmental threat has been removed. This proposed organism-level benefit would act on top of the recognized selfish gene-level selection of insertion sequences.

## MATERIALS AND METHODS

**Culturing bacterial host strains.** All *E. coli* strains (Table 1) were propagated in liquid lysogeny broth (LB) (Sigma-Aldrich: Lennox, 10 g/L tryptone, 5 g/L yeast extract, 5 g/L NaCl) at 37°C and 130 rpm shaking or statically at 37°C on lysogeny broth agar (LBA) (1.5% agar), unless otherwise stated.

**Viral enrichment—propagation of bacteriophages.** To propagate the bacteriophage isolates, a stationary phase liquid culture was diluted 1:50 and incubated at 37°C and 130 rpm until an OD$_{600}$ of 0.3 was reached. Bacteriophage stock was then added to each flask at a 1:50 dilution and samples were incubated for a further 2 h. Bacterial debris was pelleted by centrifugation at 3,220 g for 10 min before the supernatant was passed through a 0.2 $\mu$m pore size membrane filter. The crude phage stocks were stored at 4°C.

**Caesium chloride purification of bacteriophages.** The bacteriophage was propagated as described above before sodium chloride was added to a final concentration of 1 M. After incubation on ice for 1 h, samples were centrifuged at 3,220 g and the supernatant passed through a 0.2-$\mu$m pore size membrane filter before PEG8000 was added to a final concentration of 10% wt/vol. This was then left overnight at 4°C before the samples were centrifuged at 25,000 g for 60 min. The phage pellet was then re-suspended in a 6 to 7 mL SM buffer I (1 M NaCl, 8 mM MgSO$_4$ · 7H$_2$O, 25 mM Tris-HCl pH 7.5) and passed through a 0.2-$\mu$m pore size membrane filter, before undergoing concentration and further purification in a CsCl gradient (containing 1.3, 1.4, and 1.7 g/mL CsCl solutions) for 20 h at 150,000 g and 4°C. The extracted phage band was dialyzed first in SM buffer I and then twice with SM buffer II (100 mM NaCl, 8 mM MgSO$_4$ 7H$_2$O, 25 mM Tris-HCl pH 7.5). The purified phage was then stored at 4°C.

**Plaque assay—bacteriophage quantification.** The bacteriophage titer was determined via a soft agar plaque assay, using top LBA (0.7% agar) (34). One-hundred $\mu$L of serially diluted bacteriophage were incubated with 100 $\mu$L host cells (~1 × 10$^8$ CFU/mL) at room temperature for 15 min before 3 mL molten top agar was added and poured over a 90 mm 1.5% agar LBA plate and allowed to set. Plaques were quantified as PFU/mL (plaque forming units) after overnight incubation at 37°C.

For spot tests, 3 mL top agar containing only host cells was allowed to set before 5 $\mu$L of the bacteriophage of interest was spotted on top and allowed to dry before overnight incubation at 37°C.

**Twenty-four-hour growth curves.** Samples were grown in a plate reader with measurements of the OD$_{600}$ being taken every 5 min over a 24-h period. A final concentration of 1 × 10$^6$ CFU/mL *E. coli* host was added to each of the wells of a 96-well plate and grown for 4 h at 37°C with 200 rpm shaking, before bacteriophages were added to a final concentration of 1 × 10$^6$ PFU/mL. All samples were grown in LB and had a total volume of 200 $\mu$L. All growth curves were carried out using technical triplicates for biological triplicates ($n = 9$).

**Collection of phage resistant bacterial mutants.** Bacteriophage-resistant EV36 mutants were collected via streak-plating of isolates after the endpoint of triplicate 24-h growth curves exposed to K1F. Single colonies of these isolates were then cultured and assayed using growth curves and plaque assay spot tests (data not shown here). Based on the findings of these preliminary studies, two isolates showing distinctly different growth and resistance profiles were shortlisted for future study: Mutant 1 and 5 (Table 1). Mutant 1 was produced from a culture exposed to 1 × 10$^8$ PFU/mL K1F whereas Mutant 5 was created after exposure to 1 × 10$^1$ PFU/mL. Mutant 1 and 5 were then subsequently exposed to T7 and single colonies of the resistant outgrowth were again isolated from the endpoint culture from a new 24-h growth curve. Mutant 1 was the progenitor of the Mutant 1a strain and Mutant 5 was the progenitor of Mutant 5a. Removal of the K1 capsule has previously been shown to return susceptibility to the T7 bacteriophage, thereby allowing it to be used as a marker for K1 presence or absence and making it a suitable comparator for this study.

**Preparation of slides for confocal microscopy.** A stationary phase liquid culture of the bacterial host was diluted 1:100 and incubated at 37°C and 130 rpm until an OD$_{600}$ of 0.3 was reached. Cells were harvested by centrifugation and washed 3 times with phosphate-buffered saline (PBS) before resuspension in PBS to a final OD$_{600}$ of 1.0. Aliquots of 100 $\mu$L cells were then fixed for 20 min at room temperature with PBS containing 2.5% paraformaldehyde (Pierce, Thermo Fisher Scientific, cat no.10751395). After washing 3 times with 500 $\mu$L PBS, 500 $\mu$L of a blocking solution containing PBS with 3% bovine serum albumin (BSA) was added for 5 min. Cells were then centrifuged once more and re-suspended in 50 $\mu$L distilled water. Glass coverslips were cleaned with 70% ethanol, before the cell suspension was dropped on top. Coverslips were incubated at 37°C until they were completely dried ($\sim$45 min). After addition of 40 $\mu$L of primary antibody (polysialic acid recombinant monoclonal antibody (735) (Enzo, Mouse IgG2a$\kappa$, part no. ENZ-ABS560-0200), diluted 1:100 in PBS with 3% BSA), the coverslips were incubated in the fridge overnight in a sealed container.

All subsequent steps were performed protecting the samples from light. After washing coverslips 3 times with PBS, 40 $\mu$L of secondary antibody F(ab')2-Goat anti-Mouse IgG (H+L) Cross-Adsorbed Secondary Antibody, Alexa Fluor 647 (Thermo Fisher Scientific, product code: A-21237, diluted 1:200 in PBS with 3% BSA) was added and samples were incubated at room temperature for an hour. Coverslips were further washed twice with PBS and once with distilled water before being placed cell-side down on a glass slide with a drop of Fluoroshield with DAPI (Sigma-Aldrich, cat no. F6057-20ML). After being left to dry, the coverslips were then affixed to the glass slide with CoverGrip (Biotium, cat no. 23005) and stored at 4°C until required.

When working with KMS2105 (pSR647 in KMS2005; Table 1) the above protocol was carried out with the following alterations. Cultures were grown in LB with 100 $\mu$g/mL ampicillin until an OD$_{600}$ of 0.4 was reached, at this point IPTG was added to a final concentration of 0.5 mM and cultures were allowed to grow for an additional 3 h. When cells were harvested, the samples were diluted to achieve a final OD$_{600}$ of 1.0 as before and processed as previously described.

**Genomic DNA purification.** The extraction of genomic DNA for sequencing was carried out using the Invitrogen PureLink Genomic DNA minikit (Invitrogen, part no: 10053293), following the provided protocols.

**Analysis of sequencing results.** Long-read sequencing of the bacterial genomes was carried out using the Pacific Biosciences (PacBio) SMRT Platform provided by Novogene (Cambridge, United Kingdom) with a theoretical subread coverage of 286×, 289×, 268×, 875×, and 1,059× for the wild type, Mutant 1, Mutant 5, Mutant 1a and Mutant 5a, respectively. Assembly was performed with Falcon v1.8.1 (35) with error correction by Arrow v2.3.3 (36), circularization with Circlator v1.5.5 (37) and three rounds of Illumina-mediated polishing with Pilon v1.24 (38).

Complete circular genomes were annotated with Prokka v1.14.6 (39) using a priority protein reference guide of polysialic acid cluster genes and aligned with Mauve v2.4.0 (40) to compare differences.

SNPs were called with trimmed Illumina reads mapped to the EV36 wild-type reference with Snippy v4.0.2 (41). Graphical comparison of polysialic acid cluster genomic regions and the Mutant 5a deletion was performed using EasyFig v2.2.5 (42) with BLAST v2.11.0.

**Bacterial transformation.** To electro-transform plasmid pSR647 (kindly provided by Dr Willie Vann) (24), a stationary phase liquid culture of Mutant 5 was diluted 1:1,000 and incubated with shaking at 37°C until an $OD_{600}$ (optical density) of 0.4 to 0.6 was reached. The cells were centrifuged at 3,220 g for 10 min at 4°C before removing the supernatant and washing the cells twice with ice cold 10% glycerol. The washed cells were re-suspended in any residual liquid after the supernatant was removed and 80 $\mu$L of this suspension was added to 2 $\mu$L of vector. Electro-transformation was done at 2.5 kV with a 2-mm path length, before immediately adding 1 mL ice cold LB. The transformed cells were then incubated shaking at 37°C for an hour before plating onto LBA containing 100 $\mu$g/mL ampicillin and incubating statically overnight at 37°C.

## ACKNOWLEDGMENTS

We acknowledge EPSCR UKRI for the funding of this work and would like to thank Willie Vann for kindly providing the pSR647 plasmid, Dean Scholl, AvidBiotics Corporation, for providing the K1F bacteriophage, and Eric R. Vimr and Susan M. Steenbergen for providing the *E. coli* EV36 strain.

We declare that the research was conducted in the absence of any commercial or financial relationships that could be construed as a potential conflict of interest.

Growth curve analysis, mutant selection, genomic DNA purification, confocal microscopy, K.M.S.; genome analysis, R.K.L. and L.C.; writing - original draft preparation, K.M.S.; writing - review and editing, A.T.B., A.P.S., R.L., and L.C.; project administration and funding acquisition, A.T.B., A.P.S. All authors have read and agreed to the published version of the manuscript.

This research was funded by an EPSRC UKRI Innovation Fellowship, grant number EP/S001255/1. No competing financial interests exist.

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
