## [Reviewer comments · Microbiology Spectrum]

Microbiology Spectrum

Transposable element insertions into the *Escherichia coli* polysialic acid gene cluster result in resistance to the K1F bacteriophage

Kathryn Styles, Rebecca Locke, Lauren Cowley, Aidan Brown, and Antonia Sagona

Corresponding Author(s): Antonia Sagona, University of Warwick

Review Timeline:

Submission Date:	January 10, 2022
Editorial Decision:	February 4, 2022
Revision Received:	March 22, 2022
Editorial Decision:	April 4, 2022
Revision Received:	April 8, 2022
Accepted:	April 8, 2022

Editor: Adelumola Oladeinde

Reviewer(s): The reviewers have opted to remain anonymous.

Transaction Report:

DOI: <https://doi.org/10.1128/spectrum.02112-21>

February 4, 2022

Dr. Antonia P Sagona
University of Warwick
School of Life Sciences
Gibbet Hill Campus
Coventry, West Midlands CV4 7AL
United Kingdom

Re: Spectrum02112-21 (Transposable element insertions into the Escherichia coli polysialic acid gene cluster result in resistance to the K1F bacteriophage)

Dear Dr. Antonia P Sagona:

Thank you for submitting your manuscript to Microbiology Spectrum. I have reviewed your revised manuscript and your response to previous reviews. I have several concerns that need to be addressed before I can consider your manuscript for publication. When submitting the revised version of your paper, please provide (1) point-by-point responses to the issues I have raised as file type "Response to Reviewers," not in your cover letter, and (2) a PDF file that indicates the changes from the original submission (by highlighting or underlining the changes) as file type "Marked Up Manuscript - For Review Only". Please use this link to submit your revised manuscript - we strongly recommend that you submit your paper within the next 60 days or reach out to me. Detailed instructions on submitting your revised paper are below.

Link Not Available

Sincerely,

Adelumola Oladeinde

Journals Department
Editor comments:

The authors show the development of K1F phage resistance in *E. coli*. Using growth curves, microscopy, and DNA sequencing, they show that resistance is due to disruptions in the gene cluster coding for polysialic acid capsule formation. This resistance was subsequently reversed when the initial mutants were exposed to T7 phage. Sequencing suggested highly mobile insertion sequences were responsible for the gene disruptions and rearrangements.

While I agree with earlier reviews that the science seems sound for the overall study, the paper itself still needs some major revision. The introduction has references to figures and other content (for example lines 62-96) that is more appropriate for the results and discussion. The flow of the results subsections is disjointed and hard to follow. The subsections do not seem to be in a logical order. All the information and results surrounding the K1F mutants should be described prior to introducing T7. Likewise, the discussion needs re-organized and should mirror the order in the results section. Moreover, the methods should also mirror the order seen in the results. For example, section 5.2 in the methods describes one of the last experiments mentioned in the results, therefore, it should be moved towards the end of that section. Taken together, an overall re-organization would greatly improve the readability of the study and give greater clarity to its significance.

Specific line comments:

Line 12 : "This study shows how" instead of "has shown" sounds more appropriate.

Lines 62-96: The content in these sentences seem more appropriate for results and discussion as opposed to the introduction. This can be replaced with a concise paragraph that clearly says why the study was conducted and the aims of the manuscript.

Lines 66-67: What is the intended meaning of "mechanism of resistance development was highlighted via the long-read sequencing?"

Line 105: Replace "and" with "as"

Lines 108-109: Mutants 1 and 5 are not labeled on Figure 3, Panel B yet they are being individually denoted in the text. Please ensure the figures can be understood without having to read the main text. Label the mutants and make the captions clear.

Line 149 -156 - NCBI accession numbers are not working. Please release on NCBI.

Line 160: Should be Figure 7, Panel B instead of Figure 8?

Line 177: This study didn't use pathogenic bacteria so can't generalize

Line 194: Production of the capsule wasn't significantly costly to the specific strain used in this study - can't over generalize to "a bacterium."

Line 202: What is meant by, "with knock on effects?"

Lines 205-206: awkward sounding

Lines 243-244: verb tenses need adjusting

Line 250: Upregulation of what has been associated with challenges?

Figure 7, Panel B: Is this referring to DNA and genes or amino acids and proteins? Comments in figure legends and text aren't consistent. Figure 7 needs more work to improve the quality of the figure. Impossible to review due to the small size and inferior quality. Also, it states that Nanopore was used for the assembly but in the methods, there is no mention of Nanopore but Pacbio. Please clarify what sequencing platform was used.

Staff Comments:

Preparing Revision Guidelines

Please return the manuscript within 60 days; if you cannot complete the modification within this time period, please contact me. If you do not wish to modify the manuscript and prefer to submit it to another journal, please notify me of your decision immediately so that the manuscript may be formally withdrawn from consideration by Microbiology Spectrum.

Response to the Editor comments:

The authors show the development of K1F phage resistance in E. coli. Using growth curves, microscopy, and DNA sequencing, they show that resistance is due to disruptions in the gene cluster coding for polysialic acid capsule formation. This resistance was subsequently reversed when the initial mutants were exposed to T7 phage. Sequencing suggested highly mobile insertion sequences were responsible for the gene disruptions and rearrangements.

While I agree with earlier reviews that the science seems sound for the overall study, the paper itself still needs some major revision. The introduction has references to figures and other content (for example lines 62-96) that is more appropriate for the results and discussion.

Response: We thank the Editor for the very useful comments. We have now made changes in the Introduction and the content of lines 62-96 has now been distributed to results and discussion, the figures have been removed from the introduction and are now parts of the results.

The flow of the results subsections is disjointed and hard to follow. The subsections do not seem to be in a logical order. All the information and results surrounding the K1F mutants should be described prior to introducing T7.

Response: We thank the Editor for this suggestion. We have now made changes so that the subsections flow better. The figures contain data with both phage and one of the main points is that the mutants resistant to K1F also develop susceptibility to T7 phage, this is why we have selected not to describe separately the results of K1F mutants to the ones of T7 mutants. We hope that this makes sense.

Likewise, the discussion needs re-organized and should mirror the order in the results section. Moreover, the methods should also mirror the order seen in the results. For example, section 5.2 in the methods describes one of the last experiments mentioned in the results, therefore, it should be moved towards the end of that section. Taken together, an overall re-organization would greatly improve the readability of the study and give greater clarity to its significance.

Response: We thank the Editor for this suggestion. We have now re-organized the Discussion and the Methods, as suggested by the Editor.

Specific line comments:

Line 12 : "This study shows how" instead of "has shown" sounds more appropriate.

Response: this is now corrected.

Lines 62-96: The content in these sentences seem more appropriate for results and discussion as opposed to the introduction. This can be replaced with a concise paragraph that clearly says why the study was conducted and the aims of the manuscript.

Response: We thank the Editor for this suggestion. We have now removed this part and have introduced a concise paragraph as suggested.

Lines 66-67: What is the intended meaning of "mechanism of resistance development was highlighted via the long-read sequencing?"

Response: We have now clarified that.

Line 105: Replace "and" with "as"

Response: We have now corrected that.

Lines 108-109: Mutants 1 and 5 are not labeled on Figure 3, Panel B yet they are being individually denoted in the text. Please ensure the figures can be understood without having to read the main text. Label the mutants and make the captions clear.

Response: This reference has now been removed as it was confusing to the reader. This figure only shows example images of the types of plaques seen rather than being specifically of Mutants 1 or 5.

Line 149 -156 - NCBI accession numbers are not working. Please release on NCBI.

Response: We have now contacted NCBI and the NCBI accession numbers are now released. These will be fully accessible upon publication.

Line 160: Should be Figure 7, Panel B instead of Figure 8?

Response: We have now corrected that. We have changed the numbering of figures slightly, so this is now corrected.

Line 177: This study didn't use pathogenic bacteria so can't generalize

Response: We have now corrected that.

Line 194: Production of the capsule wasn't significantly costly to the specific strain used in this study - can't over generalize to "a bacterium."

Response: We have now corrected that.

Line 202: What is meant by, "with knock on effects?"

Response: We have now clarified that.

Lines 205-206: awkward sounding

Response: We have now corrected that.

Lines 243-244: verb tenses need adjusting

Response: We have now corrected that.

Line 250: Upregulation of what has been associated with challenges?

Response: We have now clarified that.

Figure 7, Panel B: Is this referring to DNA and genes or amino acids and proteins? Comments in figure legends and text aren't consistent. Figure 7 needs more work to improve the quality of the figure. Impossible to review due to the small size and inferior quality. Also, it states that Nanopore was used for the assembly but in the methods, there is no mention of Nanopore but Pacbio. Please clarify what sequencing platform was used.

Response: We have now corrected that.

April 4, 2022

Dr. Antonia P Sagona
University of Warwick
School of Life Sciences
Gibbet Hill Campus
Coventry, West Midlands CV4 7AL
United Kingdom

Re: Spectrum02112-21R1 (Transposable element insertions into the Escherichia coli polysialic acid gene cluster result in resistance to the K1F bacteriophage)

Dear Dr. Antonia P Sagona:

Thank you for sufficiently addressing earlier comments and concerns. See below further minor comments.

Lines 45-47: The details of "this study" should be presented in the last paragraph of the Introduction. Please incorporate this sentence into the last paragraph.

Lines 57-59: explanations for using particular strains are usually given in the methods or results. While introductory information about the viruses are ok here, perhaps consider removing "making it a suitable marker for this study." Please move to the methods section.

Lines 60-63 are specific to "this study" and as addressed in comment above, would be better suited at the end of the Introduction. Please incorporate this sentence into the last paragraph.

Combine lines 45 -47, lines 60 - 63 and lines 74 - 80 to make one concise last paragraph in the introduction .

Lines 45-47: In this study we investigated the K1F bacteriophage (6) and compared it to the T7 bacteriophage (7, 8) and how the E. coli strain EV36 (9) would respond to and develop resistance or sensitivities to these lytic phages. In addition,

Lines 60-63: In this study, bacteriophage-resistant EV36 isolates were collected and characterized, through the investigation of relative growth and fitness, as well as a phenotypic analysis using confocal microscopy and the sequencing of entire genomes, to identify the causative genes in the development of phage resistance.

Lines 74-80: This study aims to offer a better understanding of the role of genetics in the competition and equilibrium between bacteria and bacteriophages, showing an insight into the development of bacterial resistances via the movement of insertion sequences (IS) through the bacterial genome. Specifically, we study the natural development of resistance and then returned sensitivity, to the K1F bacteriophage, a phage which targets the K1 capsule of pathogenic Escherichia coli. The genomic flexibility provided by IS movement appears to allow rapid adaptation to the presence versus absence of bacteriophages, removing potentially energy costly mutations once they are no longer needed.

Thank you for submitting your manuscript to Microbiology Spectrum. As you will see your paper is very close to acceptance. Please modify the manuscript along the lines I have recommended. As these revisions are quite minor, I expect that you should be able to turn in the revised paper in less than 30 days, if not sooner. If your manuscript was reviewed, you will find the reviewers' comments below.

When submitting the revised version of your paper, please provide (1) point-by-point responses to the issues I raised in your cover letter, and (2) a PDF file that indicates the changes from the original submission (by highlighting or underlining the changes) as file type "Marked Up Manuscript - For Review Only". Please use this link to submit your revised manuscript. Detailed instructions on submitting your revised paper are below.

Link Not Available

Sincerely,

Adelumola Oladeinde

Reviewer comments:

Preparing Revision Guidelines

- point-by-point responses to the issues I raised in your cover letter
- Upload a compare copy of the manuscript (without figures) as a "Marked-Up Manuscript" file.
- Each figure must be uploaded as a separate file, and any multipanel figures must be assembled into one file.
- Manuscript: A .DOC version of the revised manuscript
- Figures: Editable, high-resolution, individual figure files are required at revision, TIFF or EPS files are preferred

Please return the manuscript within 60 days; if you cannot complete the modification within this time period, please contact me. If you do not wish to modify the manuscript and prefer to submit it to another journal, please notify me of your decision immediately so that the manuscript may be formally withdrawn from consideration by Microbiology Spectrum.

Response to Editor comments

Lines 45-47: The details of "this study" should be presented in the last paragraph of the Introduction. Please incorporate this sentence into the last paragraph.

Response: We have now made these changes as suggested.

Lines 57-59: explanations for using particular strains are usually given in the methods or results. While introductory information about the viruses are ok here, perhaps consider removing "making it a suitable marker for this study." Please move to the methods section.

Response: We have now made these changes as suggested.

Lines 60-63 are specific to "this study" and as addressed in comment above, would be better suited at the end of the Introduction. Please incorporate this sentence into the last paragraph.

Response: We have now made these changes as suggested.

Combine lines 45 -47, lines 60 - 63 and lines 74 - 80 to make one concise last paragraph in the introduction.

Lines 45-47: In this study we investigated the K1F bacteriophage (6) and compared it to the T7 bacteriophage (7, 8) and how the E. coli strain EV36 (9) would respond to and develop resistance or sensitivities to these lytic phages. In addition,

Lines 60-63: In this study, bacteriophage-resistant EV36 isolates were collected and characterized, through the investigation of relative growth and fitness, as well as a phenotypic analysis using confocal microscopy and the sequencing of entire genomes, to identify the causative genes in the development of phage resistance.

Lines 74-80: This study aims to offer a better understanding of the role of genetics in the competition and equilibrium between bacteria and bacteriophages, showing an insight into the development of bacterial resistances via the movement of insertion sequences (IS) through the bacterial genome. Specifically, we study the natural development of resistance and then returned sensitivity, to the K1F bacteriophage, a phage which targets the K1 capsule of pathogenic Escherichia coli. The genomic flexibility provided by IS movement appears to allow

rapid adaptation to the presence versus absence of bacteriophages, removing potentially energy costly mutations once they are no longer needed.

Response: We have now made these changes as suggested.

April 8, 2022

Dr. Antonia P Sagona
University of Warwick
School of Life Sciences
Gibbet Hill Campus
Coventry, West Midlands CV4 7AL
United Kingdom

Re: Spectrum02112-21R2 (Transposable element insertions into the Escherichia coli polysialic acid gene cluster result in resistance to the K1F bacteriophage)

Dear Dr. Antonia P Sagona:

Thank you for the opportunity to review your article.

Your manuscript has been accepted, and I am forwarding it to the ASM Journals Department for publication. You will be notified when your proofs are ready to be viewed.

Sincerely,

Adelumola Oladeinde
Editor, Microbiology Spectrum
